# A Dyadic Approach to Cancer Care: Examining the Feasibility and Preliminary Effectiveness of a Partner-Based Exercise Intervention for Caregivers and Their Care Recipients

**DOI:** 10.3390/ijerph23010056

**Published:** 2025-12-31

**Authors:** Melanie R. Keats, Thomas Christensen, Scott A. Grandy, Ross Mason, Cory A. Munroe, Stephanie Snow, Lori Wood, Christopher Blanchard

**Affiliations:** 1School of Health and Human Performance, Division of Kinesiology, Faculty of Health, Dalhousie University, Halifax, NS B3H 4R2, Canada; grandy@dal.ca; 2Physical Activity and Cancer Lab, Dalhousie University and Nova Scotia Health, Halifax, NS B3H 1V7, Canada; thomas.christensen@nshealth.ca; 3Department of Medicine, Division of Medical Oncology, Dalhousie University, Halifax, NS B3H 2Y9, Canada; stephanie.snow@nshealth.ca (S.S.); lori.wood@nshealth.ca (L.W.); 4Beatrice Hunter Cancer Research Institute, Halifax, NS B3H 0A2, Canada; 5Department of Urology, Faculty of Medicine, Dalhousie University, Halifax, NS B3H 1Y6, Canada; rossj.mason@nshealth.ca; 6Department of Psychology and Neuroscience, Dalhousie University, Halifax, NS B2H 4R2, Canada; cory.munroe@dal.ca; 7Department of Medicine, Faculty of Medicine, Dalhousie University, Halifax, NS B3H 2Y9, Canada; chris.blanchard@dal.ca

**Keywords:** family caregivers, care recipient, dyad, exercise, intervention

## Abstract

Despite being key partners in the supportive care of the cancer patient, family caregivers are often inadequately prepared for or supported to take on this critical role, subsequently putting their own wellbeing at risk and, by extension, that of the patient. Exercise interventions show promise in mitigating caregiver burden and improving health outcomes for both caregivers and patients; however, the interrelationship between family caregiver and care recipient has gone largely unexplored. Thus, we conducted a single-group pilot study to examine the feasibility and preliminary effectiveness of a 12-week dyadic exercise intervention. Of the 27 caregiver–patient dyads who consented, 21 (77.8%) completed the study, with participants completing an average of 23.1 (96.3%) of the prescribed exercise sessions, suggesting good adherence and study retention. All participants reported higher post-intervention levels of strenuous physical activity (*p* = 0.017), improved sit-to-stand repetitions (*p* = 0.004), and social functioning (*p* = 0.030) compared to baseline. Of note, caregivers reported higher post-intervention scores on the burden in relationship subscale of the Zarit Burden Interview relative to baseline (*p* = 0.043), suggesting an increase in perceived caregiver burden following the intervention. Overall, dyadic exercise interventions appear feasible and may confer certain physical and psychosocial benefits for both family caregivers and care recipients. However, such programs may also unintentionally exacerbate certain aspects of caregiver burden. Future research should aim to identify factors within dyadic exercise interventions that may contribute to increased caregiver burden, as well as strategies to help mitigate these effects.

## 1. Introduction

Recent projections suggest that, between 2024 and 2050, the global incidence of new cancer cases could increase by as much as 61% [1]. Escalating healthcare costs driven by the rising global cancer burden, as well as prioritizing patient-centered care, underscores the escalating demand for effective home-based care [2]. The shift from in-patient to out-patient care at home has expanded the responsibilities of informal, unpaid caregivers who are most often untrained family members and friends [3,4,5].

Unpaid, informal caregivers play a crucial role in supporting individuals living with cancer. Caregiving often carries the responsibility of managing several competing demands including providing emotional support, symptom management, medication administration, assisting with personal care and daily activities, transportation to and from medical appointments and helping to navigate complex healthcare systems. The contributions of caregivers are instrumental in alleviating healthcare system burdens and providing personalized and compassionate support within their communities [3,4,5]. However, these responsibilities are mentally, physically, and emotionally demanding, and are often performed over an extended period of time [4,5].

Cancer has been characterized as a family affair, as it is a disease in which both the patient and their family members are confronted by considerable physical and psychological stressors [6]. While the effects of a cancer diagnosis and its associated treatments on patient well-being have garnered much attention in recent years, the impact on the informal caregiver has received less attention, despite their significant role in cancer care. Evidence suggests that the informal caregiving role can have a negative impact on the caregivers’ overall health and well-being. Similar to the cancer patient, the informal caregiver can experience a multitude of physical (e.g., insomnia, fatigue), emotional (e.g., distress, depression, anxiety), and social consequences (e.g., relationship functioning), leading to a reduced overall quality of life and a potential increased risk of all cause-mortality [7,8,9]. It has also been reported that the unmet emotional, cognitive, or physical needs of the informal caregiver can impact their ability to manage the cancer patients’ physical, emotional, and medical (e.g., medication management) needs [5,10]. Of note, the physical and emotional health outcomes of patients and their caregivers are often found to be related. For example, high emotional stress in a spousal informal caregiver, if not treated, has been found to negatively affect the ability of the cancer patient to adjust to their diagnosis of cancer [5,10]. Regrettably, despite being a key partner in the supportive care of the cancer patient, informal caregivers are often inadequately prepared or supported to take on this critical role, subsequently putting their own health and well-being at risk [4,5,9] and, by extension, that of the cancer patient/survivor [5,6].

With a growing understanding of the significance of the informal caregiver, there has been an increase in research involving interventions designed to mitigate caregiver burden and the subsequent negative outcomes. To date, these interventions have been largely psycho-educational, focusing on increasing caregiver knowledge and skills pertaining to caregiving tasks, coping skills, and self-care [5,11,12]. To date, relatively few have specifically targeted exercise as a self-management strategy and a means of improving caregiver health and well-being, a much-needed management strategy to help caregivers attain and maintain the mental and physical stamina required to care for the cancer patient [13].

The value of exercise has been well-demonstrated, showing that virtually everyone can benefit from moving more [14,15]. Preliminary data suggest that exercise interventions show promise in mitigating caregiver burden and improving health outcomes for both the patient/survivor and caregiver [12,16]. By working together, the needs of both the patient and caregiver can be addressed through mutual social, emotional and physical support. Dyadic exercise overcomes potential barriers to exercise (i.e., time, caregiver guilt) and provides a shared experience, strengthening the relationship and reducing isolation, allowing both the patient and caregiver to better cope with and manage the disease together [12,17]. By working together, the needs of both the patient and caregiver can be addressed through mutual social, emotional and physical support, ultimately driving improved health outcomes and reduced caregiver burden [12,18]. To date however, few studies have explored the benefits of a partner-based, dyadic exercise intervention to improve health outcomes of both the cancer patient and caregiver [12].

As the interrelationship between informal caregiver and care recipient has gone largely unexplored, the primary purpose of this pilot study was to assess the feasibility of a partner-based 12-week exercise intervention. We also explored the preliminary effectiveness of the exercise program on caregiver burden and the physical and emotional health of both the informal caregiver and their care recipient.

## 2. Materials and Methods

### 2.1. Design

A pre–post-test design was used to assess intervention feasibility and preliminary evidence of effectiveness [19].

### 2.2. Recruitment

Participants were solicited through clinician or patient/caregiver self-referral. Study information posters were posted in outpatient cancer clinics in the local cancer center and were available on our website (www.thepaclab.com).

### 2.3. Eligibility

To be eligible participants, were (1) 18+ years; (2) able to perform low-to-moderate intensity physical activity; and (3) able to provide informed written consent in English. Patients with cancer were eligible if they (1) had a cancer diagnosis; (2) were scheduled to begin active treatment, were receiving treatment, or were within 3 months of treatment completion; and (3) were able to identify a primary caregiver willing to participate. In addition to the above, informal caregivers must (1) have been currently providing physical and/or psychological support to an adult cancer patient; (2) not have been diagnosed or treated for cancer in the past year; and (3) were not a bereaved caregiver. Following consent, additional screening for both patients and caregivers included the completion of the Physical Activity Readiness Questionnaire (PAR-Q+) [20] to determine appropriateness for beginning an exercise program. PAR-Qs were reviewed by the Clinical Exercise Physiologist (CEP) to ensure that it was safe for participants to engage in a new exercise program. Patients and caregivers were excluded from the study if they had any medical conditions that contraindicated exercise.

### 2.4. Procedures

Dyads who consented to study participation underwent a baseline assessment including the completion of the PAR-Q+, a pre-intervention questionnaire, and fitness assessment at our exercise oncology lab. Following the completion of baseline assessments, participant dyads took part in a 12-week, partner-based exercise program. Participants returned to the exercise lab at the end of the 12-week intervention to repeat the baseline measures.

### 2.5. Exercise Intervention

Exercise programming was overseen by a CEP and/or a Qualified Exercise Professional (QEP). Participant dyads had the option of completing the exercise intervention at a hospital-based exercise lab or virtually through a secure Nova Scotia Health Zoom link. Exercise programming was consistent with the exercise delivery model used in our other ongoing exercise oncology trials and included a combination of aerobic, resistance, balance, and flexibility exercises delivered in a partner-based setting twice weekly (for approximately 45–60 min/session), for 12-weeks. Participants were allotted an additional 2 weeks to complete any missed sessions up to a maximum of 24 sessions. Exercise intensity was tailored to each participants’ fitness level and ability, ensuring that the workload was safe, feasible, and challenging for both members of the dyad. Rolling recruitment was employed to facilitate study entry and access to exercise programming.

### 2.6. Outcome Measures

Participant characteristics included self-reported age, income, education, employment status, marital status, height and body weight (to calculate body mass index), comorbidities, overall health status, and relationship to care recipient. Data on cancer diagnosis, disease stage, and treatment(s) was collected by self-report.

### 2.7. Feasibility

Feasibility outcomes included variables associated with recruitment (consent rate) and study intervention (adherence to the exercise intervention (class attendance), reason for missed sessions, safety/adverse events, assessment completion rate, and study completion rate/retention). Assessment completion/missing data <10% and consent, attendance and study completion rates of ≥70% (good) and ≥90% (high) were set as markers of feasibility [19,21,22].

### 2.8. Effectiveness

Preliminary evidence of program effectiveness was assessed using measures of caregiver burden, fitness, self-reported physical activity behavior, sleep quality, mood, quality of life, and general health status.

Caregiver burden was assessed using the burden in relationship (6 items), loss of control over one’s life (4 items), and role strain (6 items) subscales of the Zarit Burden Interview (ZBI) [23]. The ZBI is a widely used measure of burden for caregivers of chronically ill patients, including cancer [24]. Scoring was conducted according to the guidelines provided by the ZBI manual [25]. Responses were scored on a 5-point Likert scale that ranged from 0 (never) to 4 (nearly always), with the sum of scores ranging from 0 to a maximum of 24 depending on the subscale (higher scores indicated higher burden). Cronbach’s alpha values indicated acceptable internal consistency for all three ZBI subscales at both baseline (burden in relationship: α = 0.82; loss of control over one’s life: α = 0.79; role strain: α = 0.84) and post-intervention (burden in relationship: α = 0.82; loss of control over one’s life: α = 0.83; role strain: α = 0.88).

A comprehensive assessment of fitness was based on the Canadian Society of Exercise Physiology’s Physical Activity Training for Health Protocol (CSEP-PATH) [26] and the Seniors Fitness Test [27] and included participant height and weight (calculated body mass index), waist and hip circumference, resting heart rate, resting blood pressure, 6-min walk test (6MWT; functional ability), hand grip strength, 30 s sit-to-stand (lower body muscular endurance), one legged stance (balance), and sit-and-reach (flexibility) [26].

Physical activity behavior was assessed using the Godin Leisure Time Exercise Questionnaire (GLTEQ). The GLTEQ is a well-validated three item questionnaire that assesses the frequency and duration of mild, moderate, and vigorous self-reported leisure-time exercise participation [28,29,30].

Sleep quality was assessed using the Pittsburgh Sleep Quality Index (PSQI) [31,32]. The PSQI is a 19-item tool that assesses usual sleep habits during the past month. The PSQI has demonstrated good psychometric properties in cancer patients and generates seven component scores (subjective sleep quality, sleep latency, sleep duration, habitual sleep efficiency, sleep disturbances, use of sleep medication, and daytime dysfunction) ranging from 0 (not at all) to 3 (three or more times/week). The component scores are summed to provide a global sleep quality score (range 0 to 21); a score greater than five indicates poor sleep quality [31,32].

Mood was assessed using the Depression, Anxiety and Stress Scale (DASS). The DASS is a validated 42-item scale that assesses levels of anxiety, depression and stress ranging from 0 (Did not apply to me at all) to 3 (Applied to me very much, or most of the time). Higher scores indicate more severe symptoms [33,34].

Health-related quality of life was assessed using the RAND 36-Item Health Survey. The RAND 36 is a generic measure of health status that is widely used across diverse populations and has shown good internal consistency and reliability [35,36,37]. The RAND 36 encompasses eight domains, including physical functioning, bodily pain, role limitations caused by physical health issues, role limitations due to personal or emotional problems, emotional well-being, social functioning, energy/fatigue, and perceptions of overall health. Physical and Mental composite scores were also calculated. Subscale and composite scores ranged from 0–100 with higher scores indicating better quality of life.

General health status was assessed using the EQ-5D-5L. The EQ-5D-5L is a valid, widely used tool that measures patient health across 5 dimensions (mobility, self-care, usual activities, pain/discomfort, and anxiety/depression) with five levels of severity (1 = no problems–5 = extreme problems). The visual analogue scale prompts respondents to assess their overall health by rating it from 0 (worst imaginable) to 100 (best imaginable), and each health profile can be associated with a utility index value anchored on a scale from 1 (perfect health) to 0 (dead) [38,39]. To calculate the EQ-5D-5L index score. the value set reported in Xie et al. [40] was used.

### 2.9. Statistical Analyses

As many Canadian adults tend to overestimate their physical activity levels [41], as recommended by Tabachnick and Fidell [42] we winsorized extreme METmin values (*z* > 3.29) for mild, moderate, and strenuous activities by recoding these outliers to be one unit higher than the next highest score within the sample. Two patients did not complete the questionnaire at post-intervention, while one caregiver did not complete the ZBI subscales at baseline. No imputation was performed for these cases. In cases where a participant completed the questionnaire but had missing item-level data for the PSQI (*n* = 13/574 component scores; 2.3%), DASS (*n* = 18/3444 items; 0.5%), or ZBI (*n* = 1/820 items; 0.1%), missing values were imputed using the mice package with predictive mean matching and five imputations [43]. For each missing value, the mean of the five imputed values was calculated, rounded to the nearest integer, and used in the final dataset.

All efficacy outcomes were assessed using linear mixed-effects models with Type III tests of fixed effects, implemented in R (v.4.3.1) using the lme4 package [44]. Satterthwaite’s degrees of freedom were used to calculate *p*-values via the lmerTest package [45]. All analyses included time (baseline, post-intervention) and role (patient, caregiver) as interacting fixed effects to examine changes in outcomes over time and to determine whether outcomes differed between patients and caregivers as a function of time. When a significant main effect of time or role was observed, pairwise comparisons were performed to examine differences between the two levels of the factor. Significant time x role interactions were followed up with pairwise comparisons examining the effect of time separately for patients and caregivers. Random effects of participant nested within dyad were included in all models to account for variance at both the individual and dyadic levels. All hypotheses were conducted using a Type I error rate of α = 0.05 and tested using two-sided tests. As this was a pilot clinical trial, *p* values and standardized effect size estimates (Cohen’s *d*) [46] for the effect of time were also reported separately for patients and caregivers.

Model residuals were assessed by comparing actual values with the predicted values, by using a histogram, and by using a Q–Q plot. Residuals were deemed reasonably normally distributed for all models. Observations with Cook’s Distance greater than 1 and residuals below Q1 − 1.5 × IQR or above Q3 + 1.5 × IQR, were flagged for further investigation [47]. All flagged data points were considered plausible. Therefore, no outliers were removed from any analysis. Notably, two patients reported not feeling well enough to complete the 6MWT at post-intervention and were therefore assigned a score of 0 for the 6MWT laps and meters outcomes at this time point. For the 6MWT rating of perceived exertion outcome, these cases were coded as missing data. As a score of 0 was considered a plausible value for 6MWT laps and meters, post-intervention data for these patients were retained in the analyses. When removing these datapoints altered the pattern of results, both sets of results are reported for transparency and to aid interpretation of 6MWT findings with and without patients who scored 0 at post-intervention.

As some participants took longer than 12 weeks to complete the exercise program, we conducted a sensitivity check for all models by including a duration variable dummy-coded as ≤12 weeks (0) and >12 weeks (1) as a covariate. This duration variable was nonsignificant in all analyses and did not alter the pattern of Group, Role, or Group × Role effects. Therefore, we reported the original models without the duration variable. All available data were included in the analysis. For a detailed breakdown of the number of participants included in each analysis, see the Appendix A.

## 3. Results

### 3.1. Participants

Recruitment and data collection ran between January 2023 and November 2024. A total of 43 patient–caregiver dyads expressed an interest in the exercise program, 28 (65.1%) consented to participate, and 21 (75%) completed the study, suggesting good accrual and retention. Participants completed an average of 23.1 (*SD* = 3.0) of the 24 total sessions (Median = 24; Minimum = 7), indicating strong program adherence (i.e., on average, participants completed 96.3% of sessions). Adherence rate did not significantly differ between patients (23.2 ± 2.2) and caregivers (23.9 ± 3.8), *t* (40) = −0.25, *p* = 0.80. No adverse events were noted. The most common reasons for not consenting to participate included not meeting study eligibility criteria, partner unwilling to participate, and lost to follow-up. Of the 27 dyads consenting to participate, 8 dyads opted for online programming (1 withdrew), 11 for in-person only (5 withdrew), and 8 selected hybrid programming (0 withdrew). Reasons for withdrawing included a lack of time (*n* = 1 dyad), a non-study related injury (*n* = 1 dyad), a non-study related illness (*n* = 1 dyad), and other (*n* = 3 dyads).

At baseline, 41 out of 42 participants (97.6%) completed the full survey and, at post-intervention, 40 out of 42 participants (95.2%) completed the survey, with “completed” defined as having completed all questionnaires in the survey. One caregiver did not complete the ZBI at baseline and two patients did not complete the post-intervention survey. All participants (100%) completed the full assessment protocol at baseline and 36 out of 42 (85.7%) completed all assessments post-intervention. Four participants did not return to the lab for fitness testing and two patients declined to complete the 6MWT. One participant declined given the recent placement of a catheter and associated discomfort; the other complained of severe abdominal pain that was later determined to be a result of recurrent disease.

Demographics for the 42 participants (21 patients, 21 caregivers) who completed the full study can be found in Table 1. In brief, males made up 61.9% of patients, while most of the caregivers were female (71.4%). Most patients and caregivers were White (patients: 100.0%; caregivers: 95.2%) and had a university/college degree or higher (patients: 85.7%; caregivers: 76.2%). A little under half of the sample was retired (patients: 42.9%; caregivers: 47.6%), while 23.8% of patients and 47.6% of caregivers worked either full- or part-time. The majority of caregivers were the patient’s spouse, with only three being non-spousal caregivers (parent, adult child, and friend).

Clinically, patients reported late-stage cancer (Stage III/IV: 47.6%) or were unsure of their cancer stage (42.9%). The most commonly reported cancer diagnoses included brain or spinal cord (23.8%), lung (14.3%), and prostate (14.3%). Patients reported multiple different treatments, with chemotherapy (52.4%) and radiation (38.1%) being the most common. Fatigue (85.7%) and pain (33.3%) were among the most commonly reported disease- and treatment-related side effects. A full breakdown of clinical characteristics of patients can be found in Table 2.

### 3.2. Participant-Reported Outcomes

ANOVA results for all participant-reported outcomes are reported in Table 3, while estimated marginal means for patients and caregivers at baseline and post-intervention are reported in Table 4.

In brief, significant main effects of role revealed that, compared to caregivers, patients reported lower quality of life across multiple domains, as well as lower total physical activity levels, independent of time. A significant main effect of time for strenuous physical activity levels revealed that, regardless of role, participants reported engaging in more strenuous physical activity post-intervention compared to baseline. Similarly, a significant main effect of time revealed that both patients and caregivers reported superior social functioning (RAND 36-Social Functioning) and mental well-being (RAND 36-Mental Composite Score) post-intervention compared to baseline. Finally, a time:role interaction was observed for the ED-5Q-5L-Pain/Discomfort subscale. Pairwise comparisons revealed that this significant interaction was driven by patients, *t* (39.1) = 2.40, *p* = 0.022, *d* = −0.44, but not caregivers, *t* (38.1) = −0.79, *p* = 0.44, *d* = 0.14, reporting lower levels of pain/discomfort at post-intervention compared to baseline. Finally, contrary to our hypothesis, a main effect of time revealed that caregivers reported greater relationship burden at post-intervention when compared to baseline.

### 3.3. Physical Function Outcomes

ANOVA results for all physical function/fitness outcomes are reported in Table 5, while estimated marginal means for patients and caregivers at baseline and post-intervention are reported in Table 6.

In brief, significant main effects of role for 6MWT-Laps, 6MWT-meters, 6MWT-RPE, 30 s Sit-to-Stand test, flexometer, and One-legged Balance (R) outcomes revealed that caregivers performed better than patients on multiple physical function assessments independent of time. Patients were also found to have larger waist circumferences than caregivers independent of time. Both patients and caregivers completed more sit-to-stand repetitions at post-intervention compared to baseline. Finally, a significant time:role interaction for resting diastolic blood pressure was observed. Pairwise comparisons revealed that this significant interaction was driven by caregivers, *t* (36.7) = 2.08, *p* = 0.045, d = −0.29, but not patients, *t* (36.7) = −1.06, *p* = 0.29, *d* = 0.15, displaying lower resting diastolic blood pressure at post-intervention compared to baseline. Notably, although no significant time:role interaction was observed, pairwise comparisons indicated that patients walked fewer meters on the 6MWT at post-intervention relative to baseline. This effect appeared to be driven by the two patients who scored 0 for this outcome at post-intervention. When these two 0 scores were excluded from analyses, the mean 6MWT performance was 524.6 m at baseline and 515.2 m at post-intervention; this change was not statistically significant, *t* (35.9) = 0.49, *p* = 0.63, *d* = −0.08.

## 4. Discussion

This study investigated the feasibility of a dyadic exercise intervention designed for informal caregivers and their care recipients, ultimately aiming to enhance the physical and emotional well-being and overall quality of life for both. Overall, the dyadic exercise intervention was demonstrated to be feasible, as evidenced by successful participant enrollment, good retention, and high intervention adherence [22] and study assessment completion rates. Similar to other studies, the prominent dyadic relationship was between the cancer patient and their spouse, while other family and friend relationships were less common. No study-related adverse events were reported.

For the EQ-5D-5L, although patients reported lower perceived health than caregivers on the Mobility and Usual Activities subscales, as well as on the Health Visual Analog Scale, both groups, on average, reported no or only slight problems across the five EQ-5D-5L health dimensions. Notwithstanding, patients reported significantly improved pain/discomfort and clinically meaningful improvements in their ratings of overall health (visual analog scale and index score) at post-intervention compared to baseline [48]. In contrast, caregivers were less likely to report significant or meaningful changes in perceived general health as assessed by the EQ-5D-5L. For the RAND 36, patients were also more likely to report poorer quality of life across several domains (physical functioning, limitations in physical health, energy/fatigue, social functioning, and general health) compared to caregivers. At post-intervention, both patients and caregivers reported greater social functioning compared to baseline. Beyond social functioning, caregiver quality of life outcomes remained relatively stable from baseline to post-intervention. In contrast, patients also reported significant improvements in pain, general health, and physical composite scores, as well as clinically meaningful improvements across almost all RAND 36 subscales, including both the physical and mental composite scores, based on a 3–5 point improvement threshold [36,49]. No changes in either sleep or mood outcomes for patients or caregivers were observed.

Interestingly, we found that the caregivers in the dyadic intervention reported greater perceived relationship burden post intervention compared to baseline. This contrasts with the findings of a recent systematic review suggesting that exercise interventions have a positive impact on caregiver burden and psychological well-being [50]. However, of the 13 studies included in the review, only two involved an exercise intervention that targeted both the caregiver and their care recipient. While it is not clear why caregivers in this study experienced larger relationship burdens following the exercise program, it is plausible that they were frustrated and/or overwhelmed for being asked to do more for their care recipient, rather than feeling that the dyadic exercise intervention was intended to benefit both. Notwithstanding, this finding is worth investigating and exploring the underlying feelings potentially leading to greater relationship burden.

Unsurprisingly, patients receiving or who had recently completed treatment reported lower physical activity levels than their caregivers throughout the study. However, regardless of role, participants reported engaging in higher levels of strenuous physical activity post-intervention, potentially demonstrating the value of the intervention in supporting physical activity during a particularly stressful time when many typically experience declines in activity levels. These findings are consistent with previous studies with patients [51], caregivers [52], and patient–caregiver dyads [53,54]. That said, the accuracy of self-report measures of physical activity is debated [41] and therefore the observed increase in reported strenuous physical activity levels should be interpreted cautiously. Moreover, increases in self-reported strenuous physical activity levels may be attributable to factors other than the exercise program, such as seasonality or natural recovery from treatment. These potential interpretations cannot be ruled out in the current study due to the lack of a control group. Furthermore, a small, insignificant decline in light and moderate physical activity levels was reported among caregivers. Thus, it is possible that caregivers may have decreased their day-to-day activity levels while completing the intervention. Notably, 47% of the participating patients reported having advanced (stage III/IV) cancer, providing further evidence that tailored exercise programs are feasible, safe and effective for cancer patients across various disease stages, particularly as no adverse events were reported [55,56].

When examining physical functioning outcomes, we found that caregivers performed better on several measures independent of time. However, both patients and caregivers demonstrated significant improvements in the 30-s sit-to-stand. Interestingly, patients completed fewer laps and walked less distance during the 6MWT post-intervention. The decline in 6MWT performance could be interpreted as suggesting that the intervention was too demanding for patients, potentially compounding cancer-related fatigue. While this interpretation is plausible, the decline in 6MWT performance appears to have been driven by two patients who declined to the 6MWT at post-intervention and therefore received a score of 0. Moreover, although not statistically significant, patients reported a clinically meaningful improvement on the energy/fatigue subscale of the RAND 36 based on a 3–5 point threshold [36,49], suggesting that fatigue levels were generally reduced among patients at post-intervention compared to baseline.

Christensen et al. [17] found that some caregivers express concern over dyadic exercise interventions, suggesting that any program that would be suitable for their care recipient wouldn’t be sufficiently challenging for themselves and, ultimately, wouldn’t be beneficial. Despite noting the importance of being physically active in maintaining their own health and well-being, the time required to care for their care recipient makes finding opportunities to exercise challenging. For those who do, some report feelings of guilt for taking time for self-care, potentially making dyadic exercise programming a more attractive and viable option.

While a large and growing body of literature has demonstrated the positive physical and psychosocial benefits of exercise for those preparing for, receiving, and recovering from cancer treatment, few have explored exercise as an intervention for informal cancer caregivers. Those that have explored exercise as a strategy to improve caregiver health have largely reported similar benefits for caregivers [50,57,58]. However, the benefits of dyadic exercise programs are less clear. Those that have explored dyadic interventions for informal cancer caregivers and their care recipients have largely reported that both can benefit from dyadic exercise interventions [12,54]. That said, results from this study highlight that patients and caregivers responded differently across several physical and emotional health outcomes. Given their differing needs, this is not surprising. However, it does suggest that a “one-size fits all” approach will likely not be effective, highlighting the need to tailor programming for each individual’s needs and goals. With the support of qualified exercise professionals, exercise programming can be tailored to meet the needs of all individuals, thus benefiting both the care provider and patient.

While this dyadic study addresses a significant gap in the literature by exploring the combined impact of exercise on both caregivers and their care recipients, it is not without limitations. Study limitations include its single-group, pre–post-test design and a relatively small and non-representative sample of the general cancer population, thus limiting generalizability and preventing us from drawing definitive conclusions. In particular, the absence of a control group is important to consider when interpreting participant-reported and physical functioning outcomes. Changes from baseline to post-intervention in these outcomes do not necessarily reflect intervention effectiveness and could instead be attributable to other factors, such as seasonality, natural recovery following cancer treatment, or placebo effects. Similarly, although both patients and caregivers improved in the 30-s sit-to-stand from baseline to post-intervention, these gains could reflect learning effects rather than effects of the exercise program itself. Such alternative explanations cannot be ruled out in a single-group pilot study, underscoring the need for future randomized controlled trials. Moreover, the patient sample exhibited considerable heterogeneity, with wide variation in diagnosis and disease stage, which may have influenced study outcomes and complicated the interpretation of observed differences.

The predominantly White, highly educated, and primarily spousal caregiver sample, limits the external validity of our findings, particularly in terms of applicability to racially diverse, lower socio-economic status, and non-spousal caregivers. Strategies and efforts to recruit and expand our reach to more diverse dyads are needed to better understand experiences across different populations. Continuing to explore virtual delivery methods may help reduce barriers, enhance access and foster greater inclusion. Additionally, the overt nature of the study could have introduced self-selection bias, as dyads who opted to participate may have been more motivated or interested in exercise, potentially affecting the applicability of the results to other dyads that are less inclined to engage in such an exercise intervention. Further, as follow-up measures were not included in the design, we are not able to assess the long-term impact of the intervention.

## 5. Conclusions

While largely unexplored in cancer care, individually tailored dyadic exercise programing appears feasible and offers patients and their caregivers an opportunity for both to benefit. Notwithstanding, the finding of increased caregiver burden requires careful consideration and additional exploration to determine the underlying reasons. With limited and mixed findings, additional research extending to more diverse dyads is needed.

## Figures and Tables

**Table 1 ijerph-23-00056-t001:** Demographics for patients and caregivers who completed the study (*n* = 21 dyads).

	Patients	Caregivers
Demographic Variable	*n* (%)	*n* (%)
Age ^a^	61.5 ± 13.9	59.8 ± 13.3
Sex		
Male	13 (61.9)	6(28.6)
Female	8 (38.1)	15 (71.4)
Marital Status		
Never Married	0 (0.0)	1 (4.8)
Living Common Law	5 (23.8)	5 (23.8)
Married	16 (76.2)	15 (71.4)
Ethnicity		
White	21 (100.0)	20 (95.2)
Other	0 (0.0)	1 (4.8)
Education		
Highschool or less	2 (9.5)	5 (23.8)
Community College/Non-University Certificate	5 (23.8)	2 (9.5)
Technical or Vocational School	2 (9.5)	0 (0.0)
University Certificate Below Bachelor’s Level	0 (0.0)	2 (9.5)
Bachelor’s Degree	4 (19.0)	6 (28.6)
Graduate Degree	7 (33.3)	6 (28.6)
Did Not Respond	1 (4.8)	0 (0.0)
Employment		
Full-time	4 (19.0)	7 (33.3)
Unemployed	2 (9.5)	0 (0.0)
Retired	9 (42.9)	10 (47.6)
Part-time	1 (4.8)	3 (14.3)
Homemaker/Caregiver	0 (0.0)	1 (4.8)
On Disability Leave	5 (23.8)	0 (0.0)
Household Income		
<$49,999	3 (14.3)	4 (19.0)
$50,000–$99,999	4 (19)	7 (33.3)
$100,000–$149,999	7 (33.3)	5 (23.8)
> $150,000–$199,999	4 (19.0)	3 (14.3)
Prefer Not to Answer	3 (14.3)	2 (9.5)

^a^ Age is expressed as mean ± standard deviation.

**Table 2 ijerph-23-00056-t002:** Patient reported clinical characteristics of patients.

Characteristic	*n* (%)
Cancer Diagnosis	
Brain or Spinal Cord	5 (23.8)
Breast	2 (9.5)
Kidney	2 (9.5)
Liver	1 (4.8)
Lung	3 (14.3)
Hodkin Lymphoma	1 (4.8)
Multiple Myeloma	1 (4.8)
Ovarian	2 (9.5)
Prostate	3 (14.3)
Other	2 (9.5)
Stage	
I	0 (0.0)
II	2 (9.5)
III	2 (9.5)
IV	8 (38.1)
Unsure	9 (42.9)
Treatment	
Chemotherapy	11 (52.4)
Radiation	8 (38.1)
Surgery	4 (19.0)
Hormone Therapy	2 (9.5)
Targeted Therapy	3 (14.3)
Other	3 (14.3)
Reported Side Effects	
Fatigue	18 (85.7)
Pain	7 (33.3)
Lymphedema	1 (4.8)
Peripheral Neuropathy	4 (19.0)
Muscle/Joint Issues	2 (9.5)
Bladder/Bowel Problems	3 (14.3)
Other	2 (9.5)

Note: Percentages may exceed 100%, as individual participants could report multiple treatments and/or side effects.

**Table 3 ijerph-23-00056-t003:** Analysis of variance output for all participant-reported outcomes.

Outcome	Term	F Value	*p*	*ηp* ^2^	Pattern of Results
General Health (EQ-5D-5L)					
Anxiety/Depression	Time	*F* (1, 34.4) = 0.40	0.53	0.01	
	Role	*F* (1, 18.1) = 0.13	0.72	0.00	
	Time: Role	*F* (1, 34.4) = 0.03	0.86	0.00	
Mobility	Time	*F* (1, 38.2) = 0.06	0.81	0.00	
	Role	*F* (1, 19.8) = 7.97	**0.011**	0.29	Patient mobility deficits > Caregiver
	Time: Role	*F* (1, 38.2) = 1.09	0.30	0.03	
Pain/Discomfort	Time	*F* (1, 37.4) = 1.41	0.24	0.04	
	Role	*F* (1, 20.0) = 0.11	0.75	0.00	
	Time: Role	*F* (1, 37.4) = 5.18	**0.029**	0.12	Patient baseline pain > post-intervention
Self-care	Time	*F* (1, 59.0) = 0.34	0.56	0.00	
	Role	*F* (1, 59.0) = 0.45	0.50	0.00	
	Time: Role	*F* (1, 59.0) = 0.45	0.50	0.00	
Usual Activities	Time	*F* (1, 39.6) = 1.60	0.21	0.04	
	Role	*F* (1, 20.4) = 6.68	**0.018**	0.25	Patient challenges with daily activities > Caregiver
	Time: Role	*F* (1, 39.6) = 0.68	0.42	0.02	
Health Visual Analog	Time	*F* (1, 38.0) = 1.55	0.22	0.04	
	Role	*F* (1, 19.9) = 15.78	**<0.001**	0.44	Caregiver overall health > Patient
	Time: Role	*F* (1, 38.0) = 1.35	0.25	0.03	
Index Score	Time	*F* (1, 30.2) = 1.39	0.25	0.04	
	Role	*F* (1, 16.3) = 3.89	0.066	0.19	
	Time: Role	*F* (1, 30.2) = 2.09	0.16	0.06	
Sleep					
PSQI Total Score	Time	*F* (1, 33.3) = 0.45	0.50	0.01	
	Role	*F* (1, 17.0) = 0.86	0.37	0.05	
	Time: Role	*F* (1, 33.3) = 0.45	0.50	0.01	
Physical Activity					
Mild	Time	*F* (1,58.5) = 0.25	0.62	0.00	
	Role	*F* (1,58.5) = 0.23	0.63	0.00	
	Time: Role	*F* (1, 58.5) = 0.42	0.52	0.00	
Moderate	Time	*F* (1, 39.3) = 0.37	0.55	0.00	
	Role	*F* (1, 20.2) = 1.19	0.29	0.06	
	Time: Role	*F* (1, 39.3) = 1.77	0.19	0.04	
Strenuous	Time	*F* (1, 39.9) = 6.24	**0.017**	0.14	Post-intervention PA > Baseline
	Role	*F* (1, 17.5) = 4.20	0.056	0.19	
	Time: Role	*F* (1, 39.9) = 1.17	0.29	0.03	
Total	Time	*F* (1, 38.6) = 3.32	0.076	0.08	
	Role	*F* (1, 18.0) = 5.16	**0.036**	0.22	Caregiver overall PA > Patient
	Time: Role	*F* (1, 38.6) = 0.00	0.97	0.00	
Quality of Life (RAND 36)					
Physical Functioning	Time	*F* (1, 36.5) = 1.29	0.26	0.03	
	Role	*F* (1, 18.7) = 35.32	**<0.001**	0.65	Caregiver physical functioning > Patient
	Time: Role	*F* (1, 36.5) = 0.75	0.39	0.02	
Limitations Physical Health	Time	*F* (1, 39.2) = 0.18	0.67	0.00	
	Role	*F* (1, 20.3) = 16.43	**<0.001**	0.45	Caregiver physical health > Patient
	Time: Role	*F* (1, 39.2) = 0.38	0.54	0.00	
Limitations Emotional	Time	*F* (1, 36.9) = 0.11	0.74	0.00	
	Role	*F* (1, 18.4) = 0.01	0.92	0.00	
	Time: Role	*F* (1, 36.9) = 0.12	0.73	0.00	
Energy/Fatigue	Time	*F* (1, 38.2) = 3.20	0.082	0.08	
	Role	*F* (1, 20.2) = 5.71	**0.027**	0.22	Caregiver energy > Patient
	Time: Role	*F* (1, 38.2) = 0.78	0.38	0.02	
Emotional Wellbeing	Time	*F* (1, 36.3) = 3.42	0.072	0.09	
	Role	*F* (1, 19.3) = 0.28	0.60	0.01	
	Time: Role	*F* (1, 36.3) = 0.01	0.91	0.00	
Social Functioning	Time	*F* (1, 35.8) = 5.10	**0.030**	0.12	Post-intervention social functioning> Baseline
	Role	*F* (1, 17.7) = 6.96	**0.017**	0.28	Caregiver social functioning > Patients
	Time: Role	*F* (1, 35.8) = 1.22	0.28	0.03	
Pain	Time	*F* (1, 36.5) = 2.46	0.13	0.06	
	Role	*F* (1, 37.5) = 0.99	0.33	0.03	
	Time: Role	*F* (1, 36.5) = 6.56	**0.015**	0.15	Patient baseline pain > Patient post-intervention
General Health	Time	*F* (1, 36.9) = 3.71	0.062	0.09	
	Role	*F* (1, 19.3) = 18.85	**<0.001**	0.49	Caregiver general health > Patient
	Time: Role	*F* (1, 36.9) = 2.78	0.10	0.07	
Physical Composite Score	Time	*F* (1, 37.0) = 2.00	0.16	0.05	
	Role	*F* (1, 19.5) = 22.29	**<0.001**	0.53	Caregiver PCS > Patient
	Time: Role	*F* (1, 37.0) = 3.02	0.090	0.08	
Mental Composite Score	Time	*F* (1, 36.7) = 4.37	**0.044**	0.11	Post-intervention MCS > Baseline
	Role	*F* (1, 18.9) = 1.54	0.23	0.08	
	Time: Role	*F* (1, 36.7) = 0.26	0.61	0.00	
Mood					
DASS Total Score	Time	*F* (1, 35.2) = 0.03	0.87	0.00	
	Role	*F* (1, 19.1) = 0.51	0.48	0.03	
	Time: Role	*F* (1, 35.2) = 1.24	0.27	0.03	
Depression	Time	*F* (1,34.9) = 0.57	0.46	0.02	
	Role	*F* (1, 18.8) = 0.71	0.41	0.04	
	Time: Role	*F* (1, 34.9) = 0.06	0.80	0.00	
Anxiety	Time	*F* (1, 33.9) = 0.64	0.43	0.02	
	Role	*F* (1, 18.8) = 4.26	0.053	0.18	
	Time: Role	*F* (1, 33.9) = 0.81	0.38	0.02	
Stress	Time	*F* (1, 35.2) = 0.08	0.78	0.00	
	Role	*F* (1, 19.2) = 0.24	0.63	0.01	
	Time: Role	*F* (1, 35.2) = 3.75	0.061	0.10	
Caregiver Burden					
Burden in Relationship	Time	*F* (1, 19.1) = 4.72	0.043	0.20	Post-Intervention burden > Baseline
Loss of Control	Time	*F* (1, 19.0) = 2.56	0.13	0.12	
Role Strain Subscale	Time	*F* (1, 19.2) = 1.25	0.28	0.06	

Note: DASS = Depression, Anxiety, and Stress Scale; EWB = Emotional Well-Being; MCS = Mental Composite Score; PA = Physical Activity; PCS = Physical Composite Score; PSQI = Pittsburgh Sleep Quality Inventory: QoL = Quality of Life. Time = baseline to post-intervention/Role = patient/caregiver; Time:Role = difference of change between patients and caregivers over time.

**Table 4 ijerph-23-00056-t004:** Estimated marginal means and standardized effect size estimates for all participant-reported outcomes.

Outcomes	Patients				Caregivers			
	Baseline	Post-Intervention	*p*	*d*	Baseline	Post-Intervention	*p*	*d*
General Health (EQ-5D-5L)								
Anxiety/Depression	1.81 ± 0.19	1.73 ± 0.20	0.58	−0.10	1.71 ± 0.19	1.67 ± 0.19	0.75	−0.05
Mobility	1.71 ± 0.16	1.56 ± 0.16	0.38	−0.21	1.10 ± 0.16	1.19 ± 0.16	0.57	0.13
Pain/Discomfort	1.81 ± 0.15	1.51 ± 0.16	0.022	−0.44	1.67 ± 0.15	1.76 ± 0.15	0.44	0.14
Self-care	1.05 ± 0.04	1.05 ± 0.04	0.95	0.02	1.05 ± 0.04	1.00 ± 0.04	0.37	−0.25
Usual Activities	1.76 ± 0.15	1.54 ± 0.15	0.16	−0.34	1.29 ± 0.15	1.24 ± 0.15	0.75	−0.07
Health Visual Analog	55.0 ± 4.5	61.9 ± 4.6	0.10	0.34	80.2 ± 4.5	80.5 ± 4.5	0.95	0.01
Index Score	0.85 ± 0.01	0.88 ± 0.02	0.039	0.39	0.88 ± 0.01	0.88 ± 0.01	0.82	−0.04
Sleep								
PSQI Total Score	5.8 ± 0.6	5.5 ± 0.6	0.60	−0.12	6.7 ± 0.6	6.7 ± 0.6	1.00	0.00
Physical Activity								
Mild	392.7 ± 80.9	401.6 ± 83.9	0.92	0.02	460.8 ± 80.9	391.4 ± 80.9	0.41	−0.19
Moderate	277.4 ± 94.6	392.0 ± 97.7	0.19	0.27	469.1 ± 94.6	426.2 ± 94.6	0.60	−0.10
Strenuous	98.6 ± 134.5	261.3 ± 141.3	0.33	0.26	265.7 ± 134.5	677.2 ± 134.5	0.014	0.67
Total	768.7 ± 223.6	1056.0 ± 232.1	0.22	0.28	1195.6 ± 223.6	1494.8 ± 223.6	0.19	0.29
Quality of Life (RAND 36)								
Physical Functioning	57.5 ± 4.2	64.0 ± 4.4	0.17	0.34	86.0 ± 4.2	86.8 ± 4.2	0.85	0.05
Limitations Physical Health	30.2 ± 8.5	36.7 ± 8.9	0.47	0.17	70.2 ± 8.5	69.0 ± 8.5	0.89	−0.03
Limitations Emotional	67.5 ± 8.4	67.4 ± 8.6	0.99	0.00	66.7 ± 8.4	69.8 ± 8.4	0.63	0.08
Energy/Fatigue	38.1 ± 4.4	45.8 ± 4.5	0.072	0.39	51.4 ± 4.4	54.0 ± 4.4	0.52	0.13
Emotional Wellbeing	70.0 ± 3.9	73.0 ± 4.0	0.24	0.17	67.5 ± 3.9	70.9 ± 3.9	0.16	0.19
Social Functioning	60.7 ± 5.7	74.6 ± 5.9	0.025	0.54	79.2 ± 5.7	83.9 ± 5.7	0.41	0.18
Pain	61.4 ± 5.2	76.8 ± 5.4	0.007	0.65	77.3 ± 5.2	73.6 ± 5.2	0.48	−0.16
General Health	41.4 ± 3.8	48.0 ± 3.9	0.018	0.38	64.0 ± 3.8	64.5 ± 3.8	0.85	0.03
Physical Composite Score	47.6 ± 4.3	56.2 ± 4.4	0.035	0.44	74.4 ± 4.3	73.5 ± 4.3	0.82	−0.05
Mental Composite Score	59.1 ± 4.6	64.8 ± 4.7	0.080	0.27	66.2 ± 4.6	69.7 ± 4.6	0.26	0.16
Mood								
DASS Total Score	18.8 ± 4.2	20.2 ± 4.2	0.52	0.08	16.8 ± 4.2	14.8 ± 4.2	0.36	−0.10
Depression	7.0 ± 1.6	6.2 ± 1.6	0.49	−0.10	5.1 ± 1.6	4.7 ± 1.6	0.72	−0.05
Anxiety	4.9 ± 1.1	5.7 ± 1.1	0.25	0.16	2.3 ± 1.1	2.3 ± 1.1	0.94	−0.01
Stress	6.9 ± 1.7	8.1 ± 1.8	0.26	0.15	9.3 ± 1.7	7.8 ± 1.7	0.12	−0.20
Caregiver Burden								
Burden in Relationship	NA	NA	NA	NA	5.3 ± 1.1	6.2 ± 1.0	0.043	0.20
Loss of Control	NA	NA	NA	NA	4.5 ± 0.7	5.0 ± 0.7	0.13	0.11
Role Strain	NA	NA	NA	NA	4.7 ± 1.0	5.3 ± 1.0	0.28	0.11

Note. *d* represents Cohen’s *d* for post-intervention-baseline for corresponding role. PSQI = Pittsburgh Sleep Quality Inventory; DASS = Depression, Anxiety and Stress Scale. EQ-5D-5L captures 5 levels of severity across 5 dimensions (mobility, self-care, usual activities, pain/discomfort, anxiety/depression); EQ-5D-5L visual analog scale rates overall health from (0 worst imaginable to 100 best imaginable); EQ-5D-5L index value ranks health from 1 perfect health to 0 death. Sleep (PSQI) scores greater than 5 indicate poor sleep quality. Quality of Life (RAND 36) scale and composite scores range from 0–100, with higher scores indicating better quality of life. DASS depression scores between 0 and 9, anxiety between 0 and 7, stress between 0 and 14, are considered normal; higher scores indicate greater symptom severity. Higher scores for caregiver burden subscales indicate greater burden.

**Table 5 ijerph-23-00056-t005:** Analysis of variance output for all physical function outcomes.

Outcome	Term	F Value	*p*	*ηp* ^2^	Pattern of Results
Heart Rate	Time	*F* (1, 37.9) = 0.37	0.55	0.00	
	Role	*F* (1, 39.9) = 0.55	0.46	0.01	
	Time: Role	*F* (1, 37.9) = 0.00	0.96	0.00	
Resting Systolic BP	Time	*F* (1, 36.9) = 0.95	0.34	0.03	
	Role	*F* (1, 20.2) = 1.36	0.26	0.06	
	Time: Role	*F* (1, 36.9) = 2.71	0.11	0.07	
Resting Diastolic BP	Time	*F* (1, 37.0) = 0.52	0.48	0.01	
	Role	*F* (1, 20.4) = 1.32	0.26	0.06	
	Time: Role	*F* (1, 37.3) = 4.97	**0.032**	0.12	Caregiver baseline BP < post-intervention
6 Min Walk Test Laps ^a^	Time	*F* (1, 37.9) = 0.37	0.55	0.00	
	Role	*F* (1, 19.9) = 7.69	**0.012**	0.28	Caregiver laps > Patient
	Time: Role	*F* (1, 38.5) = 2.51	0.12	0.06	
6 Min Walk Test Meters ^a^	Time	*F* (1, 37.4) = 1.57	0.22	0.04	
	Role	*F* (1, 19.8) = 7.49	**0.013**	0.27	Caregiver distance > Patient
	Time: Role	*F* (1, 38.0) = 2.90	0.097	0.07	
6 Minute Walk Test (RPE)	Time	*F* (1, 38.6) = 0.29	0.59	0.00	
	Role	*F* (1, 19.4) = 5.73	**0.027**	0.23	Patient RPE > Caregiver
	Time: Role	*F* (1, 39.2) = 0.12	0.73	0.00	
Combined Grip Strength	Time	*F* (1, 34.3) = 0.13	0.72	0.00	
	Role	*F* (1, 40.0) = 0.00	0.94	0.00	
	Time: Role	*F* (1, 34.3) = 1.36	0.25	0.04	
30 s Sit-to-Stand	Time	*F* (1, 34.5) = 9.32	**0.004**	0.21	Post-intervention > Baseline
	Role	*F* (1,19.2) = 5.72	**0.027**	0.23	Caregiver repetitions > Patient
	Time: Role	*F* (1,35.0) = 0.01	0.93	0.00	
Flexometer	Time	*F* (1,30.2) = 1.54	0.22	0.05	
	Role	*F* (1,20.3) = 11.16	**0.003**	0.35	Caregiver flexibility > Patient
	Time: Role	*F* (1,30.4) = 0.13	0.72	0.00	
One-legged Balance (L)	Time	*F* (1,37.1) = 0.36	0.55	0.00	
	Role	*F* (1,20.5) = 3.05	0.096	0.13	
	Time: Role	*F* (1,37.4) = 2.06	0.16	0.05	
One-legged Balance (R)	Time	*F* (1,37.7) = 3.57	0.066	0.09	
	Role	*F* (1,20.4) = 6.32	**0.020**	0.24	Caregiver balance > Patient
	Time: Role	*F* (1,38.0) = 0.04	0.84	0.00	
Body Mass Index	Time	*F* (1,36.1) = 0.19	0.66	0.00	
	Role	*F* (1,20.0) = 2.06	0.17	0.09	
	Time: Role	*F* (1,36.1) = 0.30	0.59	0.00	
Waist Circumference	Time	*F* (1,36.3) = 0.30	0.59	0.00	
	Role	*F* (1,20.1) = 4.85	**0.039**	0.19	Patient waist circumference > Caregiver
	Time: Role	*F* (1,36.4) = 0.00	0.95	0.00	
Hip Circumference	Time	*F* (1,36.1) = 3.12	0.086	0.08	
	Role	*F* (1,20.0) = 1.88	0.19	0.09	
	Time: Role	*F* (1,36.1) = 1.42	0.24	0.04	

Note: BP = Blood pressure; L = Left; R = right; RPE = Ratings of Perceived Exertion. Time = baseline to post-intervention/Role = patient/caregiver; Time:Role = difference of change between patients and caregivers over time. ^a^ Two patients scored 0 on the 6 Min Walk Test (6MWT) at post-intervention; excluding these 0 scores from analyses did not change the significance of Time, Role, or Time: Role effects for the Laps or Meters outcome.

**Table 6 ijerph-23-00056-t006:** Estimated marginal means and standardized effect size estimates for all physical function outcomes.

	Patients				Caregivers			
	Baseline	Post-Intervention	*p*	*d*	Baseline	Post-Intervention	*p*	*d*
Fitness Assessment Outcome								
Resting Heart Rate	71.5 ± 2.7	70.1 ± 2.8	0.64	−0.11	69.0 ± 2.7	67.8 ± 2.8	0.70	−0.09
Resting Systolic BP	123.8 ± 3.9	124.9 ± 4.0	0.64	0.06	132.5 ± 3.9	128.0 ± 4.0	0.072	−0.25
Resting Diastolic BP	76.9 ± 2.3	78.4 ± 2.4	0.29	0.15	82.2 ± 2.3	79.2 ± 2.4	0.045	−0.29
6 Minute Walk Test Laps ^a^	15.2 ± 0.9	13.8 ± 1.0	0.13	−0.34	16.8 ± 0.9	17.5 ± 1.0	0.50	0.15
6 Minute Walk Test Meters ^b^	524.6 ± 32.5	463.0 ± 33.5	0.044	−0.41	579.6 ± 32.5	588.8 ± 33.5	0.76	0.06
RPE For 6 Minute Walk Test	5.7 ± 0.5	6.0 ± 0.6	0.54	0.16	4.8 ± 0.5	4.9 ± 0.5	0.88	0.04
Combined Grip Strength	60.7 ± 5.4	62.8 ± 5.4	0.29	0.09	62.9 ± 5.4	61.7 ± 5.4	0.57	−0.05
30s Sit-to-Stand	13.2 ± 1.4	14.9 ± 1.4	0.035	0.26	16.2 ± 1.4	17.8 ± 1.4	0.040	0.25
Flexometer	15.8 ± 2.0	16.4 ± 2.1	0.55	0.06	23.6 ± 2.0	24.7 ± 2.0	0.25	0.11
One-legged Balance L	27.8 ± 3.6	26.1 ± 3.7	0.56	−0.10	32.0 ± 3.6	36.0 ± 3.7	0.16	0.24
One-legged Balance R	20.1 ± 3.9	25.6 ± 4.0	0.15	0.31	31.2 ± 3.9	35.6 ± 4.0	0.24	0.25
BMI	28.8 ± 1.2	29.0 ± 1.2	0.49	0.03	26.7 ± 1.2	26.6 ± 1.2	0.94	0.00
Waist Circumference	103.2 ± 3.0	102.6 ± 3.0	0.67	−0.04	95.3 ± 3.0	94.9 ± 3.0	0.74	−0.03
Hip Circumference	108.6 ± 2.5	109.9 ± 2.5	0.044	0.11	105.0 ± 2.5	105.2 ± 2.5	0.69	0.02

Note: BP = Blood Pressure; L=Left; R=Right ^a^ Two patients scored 0 on the 6 Minute Walk Test (6MWT) at post-intervention; when these two 0 scores were excluded from analyses, mean 6MWT laps walked for patients was 15.2 laps at baseline and 15.4 laps at post-intervention; this change was not statistically significant, *t* (36.4) = 0.26, *p* = 0.79, *d* = 0.05. ^b^ When the two 0 scores were excluded from analyses, mean 6MWT walking distance for patients was 524.6 m at baseline and 515.2 m at post-intervention; this change was not statistically significant, *t* (35.9) = 0.49, *p* = 0.63, *d* = −0.08.

## Data Availability

As participants did not give consent for their data to be shared, due to ethical restrictions, the data presented in this article is not readily available.

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
