# Peer review of "A Dyadic Approach to Cancer Care: Examining the Feasibility and Preliminary Effectiveness of a Partner-Based Exercise Intervention for Caregivers and Their Care Recipients"

_ijerph, 2025, doi:10.3390/ijerph23010056_

Round 1

Reviewer 1 Report

Comments and Suggestions for Authors

Overall, this is a well-executed and thoughtfully reported pilot study that adds valuable data on dyadic exercise interventions in cancer care. With some clarifications and refinement of interpretation, the manuscript will make a useful contribution to the literature.

-Retention (77.8%) and adherence (96.3%) are clear strengths, but feasibility would be stronger if the authors identified a priori benchmarks (e.g., ≥70% retention, ≥75% adherence) in the Methods and explicitly compared observed outcomes to these targets in the Results and Discussion. Additionally, recruitment reporting would benefit from providing the total number screened, eligible, approached, and declined, along with reasons for non-participation (if available), as these inform feasibility and real-world applicability.

-The sample is predominantly White, highly educated, and mostly composed of spousal caregivers. The Discussion should more clearly acknowledge how this limits external validity, particularly for racially diverse, lower SES, non-spousal, or multilingual caregiver groups. It would strengthen the manuscript to note how future research might recruit more diverse dyads and whether virtual delivery may enhance inclusion.

-Participants varied widely in diagnosis and stage, including many advanced or uncertain-stage cases. This heterogeneity can strongly influence fatigue, pain, function, and QoL trajectories, complicating interpretation of effectiveness. The Discussion should explicitly acknowledge the impact of this variability and, if appropriate, note any informal observations by clinicians regarding differences related to cancer type or stage, even without statistical subgroup analyses.

-Consider adding a brief note in Table 2 clarifying that percentages may exceed 100% due to multiple treatments/side effects reported.

Author Response

We would like the reviewer for the detailed and constructive comments regarding our manuscript entitled, “A dyadic approach to cancer: Examining the feasibility and preliminary effectiveness of a partner-based exercise intervention for caregivers and their care recipients”. We have reviewed the feedback and have responded to each of the comments raised by the reviewers. Our responses to the reviewer comments can be found below. We have also highlighted any revisions made to the original documents so they can be easily identified. Our responses are as follows.

Reviewer #1

Overall, this is a well-executed and thoughtfully reported pilot study that adds valuable data on dyadic exercise interventions in cancer care. With some clarifications and refinement of interpretation, the manuscript will make a useful contribution to the literature.

Retention (77.8%) and adherence (96.3%) are clear strengths, but feasibility would be stronger if the authors identified a priori benchmarks (e.g., ≥70% retention, ≥75% adherence) in the Methods and explicitly compared observed outcomes to these targets in the Results and Discussion.

Thank you for pointing that out. We have added benchmarks for recruitment/consent rate, retention, and adherence rates ≥70% as markers of acceptable feasibility in the study methods. We have also indicated missing data <10% as a marker of feasibility (Lines 164-166). The Results and Discussion sections have also been revised to reflect these changes (Lines 273-279; 388-390).   

Additionally, recruitment reporting would benefit from providing the total number screened, eligible, approached, and declined, along with reasons for non-participation (if available), as these inform feasibility and real-world applicability.

We appreciate this request. We have provided additional information on the number of participants who expressed interest, consented, and completed the study (Lines 273-285).

The sample is predominantly White, highly educated, and mostly composed of spousal caregivers. The Discussion should more clearly acknowledge how this limits external validity, particularly for racially diverse, lower SES, non-spousal, or multilingual caregiver groups. It would strengthen the manuscript to note how future research might recruit more diverse dyads and whether virtual delivery may enhance inclusion.

Thank you for your comment. We agree and have taken steps to emphasize how the study sample influences external validity (Lines 491-495).

Participants varied widely in diagnosis and stage, including many advanced or uncertain-stage cases. This heterogeneity can strongly influence fatigue, pain, function, and QoL trajectories, complicating interpretation of effectiveness. The Discussion should explicitly acknowledge the impact of this variability and, if appropriate, note any informal observations by clinicians regarding differences related to cancer type or stage, even without statistical subgroup analyses.

Thank you for this suggestion. We have added a statement noting that the heterogenous sample may have impacted study outcomes and the interpretation of these outcomes (Lines 488-490). Notwithstanding, our clinical partners have indicated that this wide variation in diagnoses and stage is representative of a real-world, population-based cancer center.

Consider adding a brief note in Table 2 clarifying that percentages may exceed 100% due to multiple treatments/side effects reported.

Thank you for this suggestion. We have added this clarifying point to Table 2.

Reviewer 2 Report

Comments and Suggestions for Authors

Dear Authors,

The manuscript addresses an extremely important and under-investigated area of oncology care—exercise-based interventions delivered within patient–caregiver dyads. I appreciate the effort invested in conducting this pilot study as well as the high retention rate (77.8%), which indicates good acceptability of the intervention.

However, the current version of the manuscript contains critical errors in data interpretation, selective reporting of results (reporting bias), and significant methodological limitations that have not been fully addressed. The most concerning issues are the discrepancies between subjective participant reports (self-declared increases in activity) and objective measures (decline in 6MWT performance), as well as a statistically significant increase in caregiver burden. These “negative” findings are, in fact, scientifically the most valuable aspect of the study, yet they appear to be minimized by the authors.

Below, I outline detailed concerns point by point.

Title and Abstract

The use of the word “Efficacy” in the title of a single-arm pre/post pilot study is misleading. This design does not permit causal conclusions regarding efficacy. I recommend changing the term to “Preliminary effectiveness” or replacing it with a more neutral wording such as “Changes in outcomes.”

The abstract reports only positive findings (increased activity, improved social functioning) while completely omitting important statistically significant results, namely: the increase in caregiver burden (p = .043) and the decline in patient performance in the 6MWT (p = .044). The abstract must reflect the results honestly rather than promote them. In its current form, it misleads the reader. Furthermore, stating that patients reported lower QoL “independent of time” provides little insight in the context of intervention effectiveness. In a study without a control group, the key parameter is change over time, not static between-group differences.

Introduction

The authors suggest that exercise can be beneficial, but there is no mechanistic explanation for why dyadic exercise specifically should reduce role strain or burden. It is plausible—perhaps even likely—that coordinating joint exercise sessions adds logistical stress. This perspective (e.g., interdependence theory) should be acknowledged.

You state that only four studies have examined dyadic exercise benefits. Please verify that this number is fully up to date given the rapidly evolving nature of this field.

Materials and Methods

The sample is extremely homogeneous: 100% of patients were White, and 85.7% had college/university education. This represents a major threat to external validity. These findings cannot be generalized to lower socioeconomic groups, for whom technology access (Zoom) or transportation pose much greater barriers. This must be recognized as a major limitation.

You indicate that winsorization was applied to GLTEQ scores due to known overestimation of physical activity by Canadian adults. The fact that such manipulation was necessary—combined with the lack of improvement in the objective 6MWT—casts doubt on the credibility of self-reported activity in this study.

The use of linear mixed models is appropriate for dyadic data, but interpreting a main effect of time as evidence of intervention effectiveness in the absence of a control group is a logical error. Time-related changes can stem from numerous factors (e.g., seasonality, natural recovery from chemotherapy).

Results

This section contains the most substantial issues with internal consistency.

Participants reported a significant increase in strenuous physical activity (p = .017). Simultaneously, their performance in the 6MWT declined significantly (from 524.6 m to 463.0 m, p = .044). This is alarming. It suggests either (a) substantial overestimation of activity in self-reports (undermining GLTEQ validity), or (b) physiological overreaching induced by the intervention, leading to impaired functional capacity. This cannot be ignored.

Table 4 shows an increase in caregiver Burden in Relationship from 5.3 to 6.2 (p = .043, d = 0.20). This indicates that the intervention—intended to help—actually increased relational burden. Adding a structured exercise requirement to an already overwhelmed caregiver schedule may be harmful.

Caregivers also showed decreases in light activity (d = −0.19) and moderate activity (d = −0.10), suggesting compensation: they may have completed the study sessions but then reduced their day-to-day activity, nullifying any health benefit.

 Discussion

The discussion requires major rewriting, as the current version is overly apologetic and speculative.

You state, “it is possible that the patient participants were less motivated to perform the task.” Explaining a >60 m decline in 6MWT performance simply as “lack of motivation” is unscientific and dismissive. Given that 52.4% of patients were undergoing chemotherapy, it is plausible that the intervention was too demanding and compounded cancer-related fatigue. This must be discussed as a safety concern and a signal for the need to tailor training intensity.

You mention that caregivers may have felt “overwhelmed” but offer no concrete solutions. The discussion should include a dedicated Clinical Implications section recommending against a one-size-fits-all approach and emphasizing the importance of monitoring caregiver stress during such interventions.

Furthermore, the manuscript must clearly state that these findings pertain to a highly privileged sample (White, educated, affluent—33% earning >$100k). In a more diverse population, practical barriers and caregiver burden would likely be even more pronounced.

Although the limitations section acknowledges the absence of a control group, its implications for interpretation (e.g., learning effects in sit-to-stand) should be discussed more thoroughly.

 Conclusions

The conclusions state that “dyadic exercise programming is feasible and offers patients and their caregivers an opportunity for both to benefit.” This is an overgeneralization that is not supported by the authors’ own findings. A more accurate statement would be:

“The program is feasible but may increase caregiver burden and requires careful adjustment of intensity for patients undergoing treatment.”

Conclusions must balance any potential benefits with the observed risks and limitations. Additionally, future research should aim to recruit more diverse samples to verify whether these findings extend beyond a White, highly educated population.

Summary

This study has potential but requires a complete reframing of its narrative. Rather than trying to demonstrate that the intervention “works,” the authors should highlight the valuable scientific insights that emerge from its challenges:

  • Why did caregivers experience increased burden?
  • Why did objective physical performance decline despite reported increases in exercise?

These are the questions that will meaningfully advance the field. Minimizing these findings diminishes the contribution of the manuscript.

 Language

The English language quality is strong and does not require a native speaker or extensive editing. Minor proofreading is needed to correct small punctuation or formatting inconsistencies. Language quality supports the manuscript’s clarity.

Example errors:

  • Inconsistent variable labels in tables (e.g., “PSOI” instead of “PSQI” in Supplementary Table 1).
  • Formatting issues at lines 4112 (“frustrated and / or overwhelmed”—spacing), 4082 (“Similar to others”).

Recommendation: Major Revisions

Kind regards,

Reviewer

Author Response

We would like to thank the reviewer for the detailed and constructive comments regarding our manuscript entitled, “A dyadic approach to cancer: Examining the feasibility and preliminary effectiveness of a partner-based exercise intervention for caregivers and their care recipients”. We have reviewed the feedback and have responded to each of the comments raised by the reviewers. Our responses to the reviewer comments can be found below. We have also highlighted any revisions made to the original documents so they can be easily identified. Our responses are as follows.

REVIEWER 2

However, the current version of the manuscript contains critical errors in data interpretation, selective reporting of results (reporting bias), and significant methodological limitations that have not been fully addressed. The most concerning issues are the discrepancies between subjective participant reports (self-declared increases in activity) and objective measures (decline in 6MWT performance), as well as a statistically significant increase in caregiver burden. These “negative” findings are, in fact, scientifically the most valuable aspect of the study, yet they appear to be minimized by the authors.

Below, I outline detailed concerns point by point.

Title and Abstract

The use of the word “Efficacy” in the title of a single-arm pre/post pilot study is misleading. This design does not permit causal conclusions regarding efficacy. I recommend changing the term to “Preliminary effectiveness” or replacing it with a more neutral wording such as “Changes in outcomes.”

Thank you for this comment. We have changed “efficacy” to “preliminary effectiveness” in the title.

The abstract reports only positive findings (increased activity, improved social functioning) while completely omitting important statistically significant results, namely: the increase in caregiver burden (p = .043) and the decline in patient performance in the 6MWT (p = .044). The abstract must reflect the results honestly rather than promote them. In its current form, it misleads the reader. Furthermore, stating that patients reported lower QoL “independent of time” provides little insight in the context of intervention effectiveness. In a study without a control group, the key parameter is change over time, not static between-group differences.

Thank you for these comments. We have changed efficacy to preliminary effectiveness in both the Title and the Abstract. We have also removed references to main effects of role from the abstract, as we agree that these are secondary in the context of the research questions. Finally, we have noted the increase in the burden in relationship subscale of the Zarit Burden Interview in the abstract. We now conclude the abstract by highlighting that future research should aim to identify factors within dyadic exercise interventions that may contribute to increased caregiver burden, as well as strategies to help mitigate these effects.

We did not include the 6MWT (meters) result in the updated abstract, as the significant decline in this outcome was driven by two patients who did not feel well enough to complete this assessment at post-intervention, and thus were given a score of 0. Instead, we elaborate on this result in more detail throughout the body of the text (please see below for details). We feel that this result needs to be explained in more detail than the abstract allows to avoid readers misinterpreting the nature of the effect.

Introduction

The authors suggest that exercise can be beneficial, but there is no mechanistic explanation for why dyadic exercise specifically should reduce role strain or burden. It is plausible—perhaps even likely—that coordinating joint exercise sessions adds logistical stress. This perspective (e.g., interdependence theory) should be acknowledged.

Thank you for your comment. As we found that caregivers reported greater burden post-intervention, we agree that it is possible that exercise added logistical stress. We have highlighted this finding in the Results (Lines 342-344), Discussion (Lines 411-421), and Conclusion section (Lines 504-508) We have also provided a more detailed explanation for why dyadic exercise might reduce caregiver burden in the Introduction (Lines 90-97).

You state that only four studies have examined dyadic exercise benefits. Please verify that this number is fully up to date given the rapidly evolving nature of this field.

When preparing the manuscript, we conducted a comprehensive review of the literature to explore the benefits of dyadic exercise interventions for both patients and caregivers within a cancer care context. In addressing the reviewers’ comments, we conducted an additional search and did not find any new published findings. As we did not conduct a systematic review, we have softened the language in the introduction indicating that “few studies” have explored dyadic exercise interventions for both cancer patients and their caregivers (Line 97).

Materials and Methods

The sample is extremely homogeneous: 100% of patients were White, and 85.7% had college/university education. This represents a major threat to external validity. These findings cannot be generalized to lower socioeconomic groups, for whom technology access (Zoom) or transportation pose much greater barriers. This must be recognized as a major limitation.

We agree and have taken steps to emphasize how the study sample may influence external validity (Lines 491-495).

You indicate that winsorization was applied to GLTEQ scores due to known overestimation of physical activity by Canadian adults. The fact that such manipulation was necessary—combined with the lack of improvement in the objective 6MWT—casts doubt on the credibility of self-reported activity in this study.

Thank you for this comment. We agree that the validity of self-report measures of physical activity is of potential concern, and we now highlight this point of the revised Discussion (Lines 428-430). Please see our comments below regarding associations between the 6MWT and self-reported physical activity levels.

The use of linear mixed models is appropriate for dyadic data, but interpreting a main effect of time as evidence of intervention effectiveness in the absence of a control group is a logical error. Time-related changes can stem from numerous factors (e.g., seasonality, natural recovery from chemotherapy).

Thank you. We agree that main effects of time do not function inherently as evidence of intervention effectiveness. In our original manuscript, we tried to be careful with our wording regarding this point. To more strongly emphasize that we do not interpret main effects of time to reflect evidence of intervention effectiveness, we have reiterated the lack of a control group in the revised Discussion as well as in Section 2.1 Design (Line 109) to state that, “A pre-post-test design was used to assess intervention feasibility and preliminary evidence of effectiveness”. We have also expanded the Discussion to address how the absence of a control group should be considered when interpreting observed effects (Lines 433-434; 479-488).

Results

This section contains the most substantial issues with internal consistency.

Participants reported a significant increase in strenuous physical activity (p = .017). Simultaneously, their performance in the 6MWT declined significantly (from 524.6 m to 463.0 m, p = .044). This is alarming. It suggests either (a) substantial overestimation of activity in self-reports (undermining GLTEQ validity), or (b) physiological overreaching induced by the intervention, leading to impaired functional capacity. This cannot be ignored.

Thank you for pointing this out. The significant decline in 6MWT performance among cancer patients was driven by two patients who did not feel well enough to perform the 6MWT at post-intervention and were therefore assigned a score of 0 for both the 6MWT Laps and 6MWT Meters outcomes (reasons explained in Lines 291-295). These participants were retained in the primary analyses, as a score of 0 was considered a plausible value and our default analytic approach was to retain all data deemed plausible.

When post-intervention 6MWT data from these two patients were excluded, 6MWT performance among cancer patients declined only from 524.6 m at baseline to 515.2 m at post-intervention (Table 6). This change was not statistically different from zero, t(35.9) = −0.49, p = .63, and was small in magnitude (d = −0.08). Similarly, with these two patients excluded, patients completed an average of 15.2 (SE = 0.8) laps at baseline and 15.4 (SE = 0.8) laps at post-intervention; this change was also not statistically significant, t(36.4) = 0.26, p = .79, d = 0.05.

We acknowledge that this additional context is important to fully interpret the 6MWT results. Accordingly, the revised manuscript now reports results both including and excluding the two patients who scored 0 at post-intervention for the 6MWT Meters outcome (Tables 5 and 6; Lines 376-382.

To further explore the relation between self-reported physical activity and objective functional performance, we examined correlations between changes in self-reported total physical activity and changes in 6MWT performance among cancer patients. Changes in self-reported total physical activity were positively associated with changes in 6MWT performance (meters: r = .32, p = .21; laps: r = .33, p = .19). Although these correlations did not reach statistical significance, they suggest that patients who reported greater increases in total physical activity levels tended to show greater improvements in 6MWT performance, providing some support for the validity of the self-report physical activity measures (though we acknowledge that such measures are not perfect).

Table 4 shows an increase in caregiver Burden in Relationship from 5.3 to 6.2 (p = .043, d = 0.20). This indicates that the intervention—intended to help—actually increased relational burden. Adding a structured exercise requirement to an already overwhelmed caregiver schedule may be harmful.

Thank you for this comment. We agree that this is a plausible interpretation of the result and have noted that in the manuscript (Lines 411-421).

Caregivers also showed decreases in light activity (d = −0.19) and moderate activity (d = −0.10), suggesting compensation: they may have completed the study sessions but then reduced their day-to-day activity, nullifying any health benefit.

Thank you for this comment. We agree that this is a plausible interpretation of the results and have therefore added this point to the revised Discussion in paragraph (Lines 434-437).

Discussion

The discussion requires major rewriting, as the current version is overly apologetic and speculative. You state, “it is possible that the patient participants were less motivated to perform the task.” Explaining a >60 m decline in 6MWT performance simply as “lack of motivation” is unscientific and dismissive. Given that 52.4% of patients were undergoing chemotherapy, it is plausible that the intervention was too demanding and compounded cancer-related fatigue. This must be discussed as a safety concern and a signal for the need to tailor training intensity.

Thank you for your observation. We have revised the Discussion section to highlight the clinically meaningful improvements in the Energy/Fatigue subscale of the RAND 36, suggesting that the intervention did not worsen cancer-related fatigue (Lines 449-452). We have also highlighted that patients reported clinically meaningful improvements across most of the RAND 36 subscales as well as some EQ-5D-5L measures. While these are not direct measures of fatigue, it seems reasonable to assume that these self-reported measures would not improve should the exercise program be deemed too demanding or to have worsened fatigue. As above, we have also highlighted the reason why the 6MWT results were lower as 2 patients did not perform the post-intervention walk.

You mention that caregivers may have felt “overwhelmed” but offer no concrete solutions. The discussion should include a dedicated Clinical Implications section recommending against a one-size-fits-all approach and emphasizing the importance of monitoring caregiver stress during such interventions.

Thank you. We have expanded the study conclusions highlighting the need to individualize exercise prescriptions and the importance of exploring the potential reasons for increased caregiver burden (Lines 468-474; 504-508).

Furthermore, the manuscript must clearly state that these findings pertain to a highly privileged sample (White, educated, affluent—33% earning >$100k). In a more diverse population, practical barriers and caregiver burden would likely be even more pronounced.

We agree and have added additional explanation in the study limitations (Lines 491-495).

Although the limitations section acknowledges the absence of a control group, its implications for interpretation (e.g., learning effects in sit-to-stand) should be discussed more thoroughly.

Thank you for this comment. We agree that it is critical to consider the lack of a control group when interpreting the effects observed in the current study. We have expanded on this point in the revised Discussion (Lines 479-488).

 Conclusions

The conclusions state that “dyadic exercise programming is feasible and offers patients and their caregivers an opportunity for both to benefit.” This is an overgeneralization that is not supported by the authors’ own findings. A more accurate statement would be:

“The program is feasible but may increase caregiver burden and requires careful adjustment of intensity for patients undergoing treatment.”

Conclusions must balance any potential benefits with the observed risks and limitations. Additionally, future research should aim to recruit more diverse samples to verify whether these findings extend beyond a White, highly educated population.

Thank you for those comments. We have revised the Conclusions to better reflect the study limitations and findings (Lines 504-508).

Summary

This study has potential but requires a complete reframing of its narrative. Rather than trying to demonstrate that the intervention “works,” the authors should highlight the valuable scientific insights that emerge from its challenges:

Why did caregivers experience increased burden?

While it is unclear why caregivers experienced increased burden, we have speculated that the additional expectations and potential perceived need to more for the care recipient led to a feeling of higher burden (Lines 416-421).

Why did objective physical performance decline despite reported increases in exercise?

We have addressed the apparent inconsistencies in physical performance (i.e., 6MWT) in a previous comment.

These are the questions that will meaningfully advance the field. Minimizing these findings diminishes the contribution of the manuscript.

We thank the reviewer for the insightful and detailed feedback.

 Language

The English language quality is strong and does not require a native speaker or extensive editing. Minor proofreading is needed to correct small punctuation or formatting inconsistencies. Language quality supports the manuscript’s clarity.

Example errors:

  • Inconsistent variable labels in tables (e.g., “PSOI” instead of “PSQI” in Supplementary Table 1).

Thank you. We have reviewed the tables for consistency.

  • Formatting issues at lines 4112 (“frustrated and / or overwhelmed”—spacing), 4082 (“Similar to others”).

Thank you for catching this. This has been corrected in the revised manuscript.

Round 2

Reviewer 2 Report

Comments and Suggestions for Authors

The Authors have adequately addressed all major concerns raised in the previous round of review. The manuscript now meets the standards for publication as a transparent report of a pilot study. I have no further comments.